# MEQA: A Benchmark for Multi-hop Event-centric Question Answering with Explanations

**Ruosen Li**, **Zimu Wang**, **Son Quoc Tran**, **Lei Xia**, **Xinya Du**

Department of Computer Science, University of Texas at Dallas
`{ruosen.li, zimu.wang, lei.xia, xinya.du}@utdallas.edu`

## Abstract

Existing benchmarks for multi-hop question answering (QA) primarily evaluate models based on their ability to reason about entities and the relationships between them. However, there's a lack of insight into how these models perform in terms of *both events and entities*. In this paper, we introduce a novel semi-automatic question generation strategy by composing event structures from information extraction (IE) datasets and present the first Multi-hop Event-centric Question Answering (MEQA) benchmark[1]. It contains (1) 2,243 challenging questions that require a diverse range of complex reasoning over entity-entity, entity-event, and event-event relations; (2) corresponding multi-step QA-format event reasoning chain (explanation) which leads to the answer for each question. We also introduce two metrics for evaluating explanations: completeness and logical consistency. We conduct comprehensive benchmarking and analysis, which shows that MEQA is challenging for the latest state-of-the-art models encompassing large language models (LLMs); and how they fall short of providing faithful explanations of the event-centric reasoning process.

## 1 Introduction

Multi-hop Question Answering (QA) is an important task that challenges NLP models' ability, including large language models (LLMs), to perform multi-step reasoning to answer the given question based on pieces of information (and their relationships) from the context [Yang et al., 2018; Mavi et al., 2022]. The ability to conduct multiple reasoning steps is important because it empowers models to understand and perform complicated real-world tasks that require information aggregation in/across documents. Examples include sentence fusion [Brook Weiss et al., 2022], multi-document summarization [Haghighi and Vanderwende, 2009], timeline summarization [Yan et al., 2011], and event logical/temporal reasoning [Yang et al., 2020, 2023, 2024a,b].

However, current research predominantly focuses on entity-centric questions, leaving event structure and event-event relations underrepresented in semantic sources and corresponding QA datasets [Souza Costa et al., 2020]. For instance, Figure 1 illustrates an event-centric question encompassing both entities and events, a domain often overlooked by popular QA benchmarks [Ho et al., 2020; Trivedi et al., 2022]. Event-centric questions are inherently more complex than entity-centric ones, as discussed in the subsequent section. They demand a compositional understanding of both entity and event knowledge, presenting significant challenges for NLP models, including LLMs, and serving as a robust benchmark for evaluating reasoning capabilities [Souza Costa et al., 2020].

Motivated by this gap in the QA and machine reading comprehension literature, we present the first Multi-hop Event-centric Question Answering (MEQA) dataset. It includes 2,243 questions, requiring

---

[1]Our benchmark is publicly available at `https://github.com/du-nlp-lab/MEQA`.

38th Conference on Neural Information Processing Systems (NeurIPS 2024) Track on Datasets and Benchmarks.

> **Document**:
> […] nation's Defense Ministry confirmed that a **major general** was **killed** in Syria by an improvised explosive device, *Al-Monitor* online *reported*. […] In 2017, a **lieutenant general** was **killed** in the same province, […]
> **Q**: Who **died** before *Al-Monitor* reported online?
> **A**: **major general**, **lieutenant general**

Figure 1: An example of multi-hop event-centric question in MEQA. Models should start reasoning from the *Al-Monitor* and first locate the *reported* event; then find all **events** that happened before the reported event; and finally extract **victims** in all those events, which are answers to the question.

a diverse range of reasoning capabilities to answer, for example, event relations, entity bridging, and event listing and counting. In order to bootstrap the annotation process for collecting multi-hop questions, we propose re-purposing the existing information extraction (IE) dataset (i.e., WikiEvents [Li et al., 2021]), which includes annotated event structures (event triggers and entities). Specifically in our work, we design a novel question generation strategy: firstly, examine document-level event structures, followed by linking events into event reasoning chains in the given document, and finally, generate synthetic questions and QA-pair style explanations. Human annotators later curate the synthetic questions and explanations.

The key contributions of our work can be summarized as follows: (1) we collect the first challenging multi-hop event-centric question answering dataset with explanations (MEQA): Our empirical findings underscore the uniqueness of our benchmark that presents novel challenges and a wide range of diversity. Notably, our results reveal a substantial gap between the performance of state-of-the-art language models and human performance, which provides a promising avenue for future research; (2) we introduce a bottom-up process that partially automates the dataset construction process by identifying composable events from the IE dataset; (3) we introduce the completeness and logical consistency metrics to evaluate generated explanations, which are efficient and align well with human judgments; (4) we propose methods that leverage informative structured information (e.g., entity and event) for our new task, which significantly improves performance and generates a more faithful reasoning process.

## 2   Related Work

**Multi-hop QA.** Previous multi-hop QA benchmarks all focus on entity-relation understanding [Das et al., 2019; Saxena et al., 2020; Fang et al., 2020]. HotpotQA [Yang et al., 2018] is constructed with a top-down approach by directly crowdsourcing multi-hop questions, which is later shown to be solvable using single-hop shortcuts [Chen and Durrett, 2019]. On the other hand, 2WikiMultihopQA [Ho et al., 2020] and Musique [Trivedi et al., 2022] are constructed using the bottom-up approach, where multi-hop questions are composed of single-hop reasoning steps. While they all rely on initially human-written questions, MEA-QG [Pan et al., 2021] generates multi-hop questions by first selecting relevant information from different data sources with a set of operators and then integrating the multiple information to form a question with six reasoning graphs. Our MEQA is constructed with a novel bottom-up approach to bootstrap question generation (QG), as we use the labels from event extraction datasets for identifying composable single-hop steps.

**Explainable Complex QA.** Datasets for QA that feature intricate reasoning questions frequently include comprehensive explanations. These explanations serve the dual purpose of allowing QA systems to learn with stronger supervision and facilitating evaluation of models' ability to explain their predictions [Li and Du, 2023a; Du, 2024; Li et al., 2024]. HotpotQA [Yang et al., 2018] only includes evidence sentences from the passages. StrategyQA [Geva et al., 2021] includes question decomposition results, but they do not serve as explanations. ScienceQA [Lu et al., 2022] includes free-form explanations for scientific questions. Our MEQA focuses on the most natural free-form explanations and includes single-hop QA pairs as the reasoning path (explanations) that leads to the final answers.

**Event-centric QA.** Simultaneous with the research progress in event understanding, i.e., event extraction (EE) and event relation extraction (ERE) [Du and Cardie, 2020; Du and Ji, 2022; Wang et al., 2022, 2024; Mehta et al., 2022; Peng et al., 2023a,b; Jin et al., 2023; Choudhary and Du, 2024], multiple event-centric QA datasets have been proposed with different characteristics. EventQA

Table 1: Types of complex event-based questions in our MEQA dataset. We highlight the answer and its supporting facts in **bold texts**, event triggers in *blue*, bridging entities in *green*. In the **Event Comparison** type, *blue* and *green* texts only represent related content for two compared entities.

| Type | Context | Question | Answer | Explanation |
|---|---|---|---|---|
| ***Event Relation*** (49.7%) | [...] a ***major general*** was *killed* [...] ***Al-Monitor*** online reported. [...] A ***local commander*** [...] was also reportedly *killed* [...] a ***lieutenant general*** was *killed* [...] | Who died before ***Al-Monitor*** reported it online? | major general, local commander, lieutenant general | 1. What event contains ***Al-Monitor*** as the communicator? *reported* 2. What events in #1 contain victims? *kills*, *killed* 3. Who are the victims in #2? ***major general***, ... |
| ***Entity Bridging*** (37.1%) | [...] in *Belfast*, at least ***11 people*** died. [...] Early today Mr Whitelaw came back to *Belfast* by ***plane*** [...] | Which transportation method did Whitelaw use to reach the place where ***11 people*** died? | Plane | 1. Where did ***11 people*** die in explosion? *Belfast* 2. Which transportation method did Whitelaw use to reach the #1? ***Plane*** |
| ***Event Listing and Counting*** (6.2%) | Roadside IED *kills* Russian ***major general*** in Syria. [...] ***Three military personnel*** were *wounded*, [...] A ***local commander*** [...] was also reportedly *killed* [...] | How many victims are mentioned in the whole text? | 5 | 1. What events contain victims? *kills*, *wounded*, *killed* 2. How many victims are in #1? ***5*** |
| ***Event Comparison*** (5.0%) | Roadside ***IED*** kills Russian *major general* in Syria. [...] *Three military personnel* were wounded, [...] In April, *two dozen Syrian fighters* were killed in *IS* attacks [...] | According to the document, which one, ***IED*** or ***IS***, killed more victims? | IS | 1. Who was killed by ***IED***? *major general, three military personnel* 2. Who was killed by ***IS***? *two dozen Syrian fighters* 3. Which one between #1 and #2 killed more victims? *IS* |
| ***Unanswerable*** (2.0%) | Early today Mr Whitelaw [...] 21 bombs had gone off [...] Mr Whitelaw and Lord Carrington immediately flew back [...] | Who traveled to the place where 21 bombs were manufactured? | (No Answer) | - |

[Souza Costa et al., 2020], utilizes a random walk on the EventKG [Gottschalk and Demidova, 2019], is designed for accessing semantic data stored within KGs; however, it focuses on KG and is not multi-hop. TORQUE [Ning et al., 2020] includes questions querying commonsense temporal relationships based on the MATRES dataset [Ning et al., 2018]. ESTER [Han et al., 2021] focuses on the challenges of five semantic relations between events. Our MEQA dataset differs from above and it includes event-centric questions that require multi-hop reasoning, namely, temporal/causal relation identification, event trigger-argument entity, and argument-argument understanding.

## 3 Complex Event-centric Questions

### 3.1 Question Strategies

Following the definition in ACE (Automatic Content Extraction) [Walker et al., 2006] of entities, relations, and events, we define our event-centric questions in Appendix A. Table 1 provides a breakdown of question types within our dataset and the corresponding proportions. To increase diversity, we follow five strategies to annotate multi-hop questions, with examples in Appendix B:

**Event Relation.** In this type of question, language models are tested on the ability of determining the relation between different events, including (1) event-event relation: encompassing causal relations

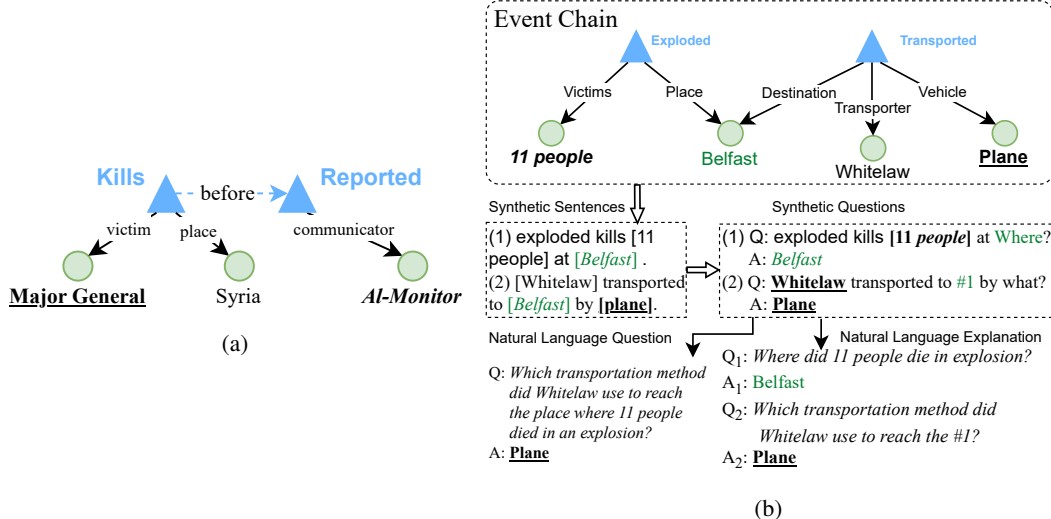

Figure 2: Partial graph for the Strategy 1 (a) and Strategy 2 (b) examples. The middle part of (b) displays the synthetic sentences and explanations. The bottom part of (b) shows the finalized natural language question and explanation.

(e.g., CAUSE) and temporal relations (e.g., BEFORE). (2) event-entity relation: relations between events and entities are represented by the role of entities. Figure 2(a) is an example of the strategy.

**Entity Bridging.** This type of multi-hop question tests the connected reasoning ability [Trivedi et al., 2022]. The event reasoning chains connect or bridge events from different parts of the documents by entities. They only contain one type of relation: event-entity relation. Events are connected by bridging entities but not direct event-event relations. Figure 2(b) displays an example of the strategy.

**Event Listing and Counting.** This type of question tests the discrete reasoning skills of language models [Dua et al., 2019]. In the third example in Table 1, models should first follow similar steps in Strategy 1, including finding all related events and extracting candidate entities, and then calculating the total amount of victims.

**Event Comparison.** This type of question also tests the discrete reasoning skills of language models [Dua et al., 2019]. For example, in the fourth sample in Table 1, language models should count the five people killed by the IED and the 24 people killed by IS, and then make the correct comparison that the number of victims by IS (24) is greater than the number of victims by the IED (5).

**Unanswerable Questions.** Questions of this type are those that cannot be answered based only on the corresponding documents. In this question type, we provide annotators with a list of already annotated answerable questions and encourage them to write unanswerable questions that closely resemble the answerable ones.

## 3.2 Multi-hop Event Reasoning Desiderata

**Multi-hop Reasoning.** Multi-hop event reasoning questions require language models to comprehend the connections between various events within the provided text. Each question entails a multi-step reasoning process, which can be broken down into a sequence of simpler inquiries. A straightforward question can be resolved by (1) referencing a brief passage within the document or (2) employing logical operations based on information gathered from prior steps.

For instance, consider the explanation of the "Event Comparison" strategy outlined in Table 1. The initial two steps can be addressed by referring to concise text excerpts within the document. However, to answer the final question, language models must apply logical operations to the information previously acquired.

**Reasoning Shortcuts** pose a significant challenge to the quality of multi-hop reasoning datasets [Min et al., 2019; Trivedi et al., 2022]. Suppose a question is built based on two events $e_1$ and $e_2$, in which $e_1$ starts the entire reasoning chain and $e_2$ contains the final answer. If $e_2$ can be directly and uniquely located in the context by a combination of its trigger and a set of relations, $e_1$ is no longer required. For example, the TRANSPORTED event in Figure 2(b) can be located in the document if we

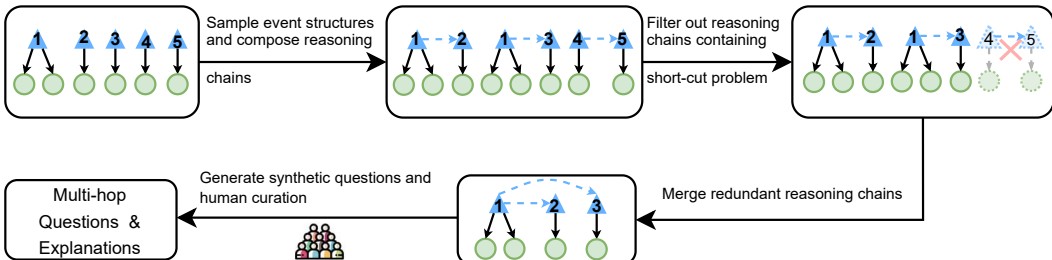

Figure 3: MEQA construction pipeline includes identifying composable events from any IE dataset, composing reasoning chains, mitigating short-cut problems, merging similar reasoning chains to reduce redundant and incomplete questions and, generating synthetic multi-hop questions and QA-based explanations. MEQA pipeline also employs human curation to finalize the high-quality dataset.

assume "plane" only appears once. The second stage of our data collection pipeline, Section 4.2, is specifically designed to eliminate the potential reasoning shortcuts in our dataset.

# 4 Data Collection Pipeline

As illustrated in Figure 3, the construction of MEQA involves four main steps: (1) identifying and linking composable events from IE datasets to event reasoning chains; (2) removing disqualified and merging redundant reasoning chains; (3) converting reasoning chains to synthetic questions and explanations; and (4) human curations.

## 4.1 Composing Reasoning Chains from Event Structures

We first leverage the event structures from the WikiEvents [Li et al., 2021] (Appendix C) dataset to find composable events (e.g., events that share an entity or have relations in between) by different strategies. The schema of WikiEvent (KAIROS, see Appendix D) describes different types of events (e.g., TRAVEL) and their corresponding argument roles (e.g., TRANSPORTER and ORIGIN). For example, "[TRANSPORTER] traveled from [ORIGIN] to [DESTINATION]".

The first step is to filter composable events and link them together to form event reasoning chains, which are event reasoning chains in the QA task. For **Strategy 1**, we randomly select a pair of events, regard one of them as the start event and the other one as the end event. Then, we prompt LLM to generate a temporal or causal relationship between them (Appendix G). For **Strategy 2**, we randomly pick one event as the start event. Then, we search for another event, sharing at least one common entity with the start event (e.g., the EXPLODED event in Figure 2(b)), and concatenate them to form a reasoning chain. Currently, this is a two-hop reasoning chain, and the second event is the end event (e.g., the TRANSPORTED event in Figure 2(b)). For more hops reasoning chains, we repeatedly add more composable events to the end of the reasoning chain. The maximum hop is 4 in this strategy. For the rest of the strategies, we ask annotators to select events manually before writing down questions.

## 4.2 Relieving Reasoning Shortcut Problem

As described in Section 3.2, uniquely located events lead to reasoning shortcut problems. The primary approach to address this involves ensuring that all event structures, except for the start event, cannot be uniquely identified solely by their triggers and chosen argument roles.

**Automatic Process.** For Strategy 1, during the event sampling process, we record a warning in the data if an end event is unique so annotators can notice the problem and modify it during the annotation process. For event structures used to form an event reasoning chain in Strategy 2, we only keep arguments that connect two events and the corresponding roles and leave out the others. If this minimum event structure can still be uniquely located in all events, we disregard the event and select another one for the current reasoning chain. We keep the reasoning chains containing at least two hops. For other strategies, annotators mitigate the problem by mimicking the above process.

**Manual Correction.** For Strategy 1, annotators remove arguments in the end event to ensure it is not unique in all events (it does not indicate the answer is not unique). The reasoning chain should be discarded if it cannot meet the requirement. If the start event is not unique, annotators will add more

information from the document to the event structure to ensure it can be uniquely located. It is the same for start events in all strategies.

### 4.3 Merging Redundant Reasoning Chains

When constructing event reasoning chains, many reasoning chains share most of the same event structures except the end event, which includes the answer of the corresponding chain. We merge them together to reduce redundant reasoning chains and combine answers to the same chains. For Strategy 1, each reasoning chain only contains two events. If two event chains have the same start event and event-event relation, we can merge the two chains, as shown in Figure 3. Suppose we have two reasoning chains: the WOUNDED event happened before the REPORTED event, and so did the KILLED event. As REPORTED is the start event in both reasoning chains, we can merge them.

### 4.4 Generating Synthetic Question Answer Pairs and Explanations

Our goal is to generate multi-hop synthetic questions and explanations based on event reasoning chains. Firstly, we utilize (1) the descriptions of events and argument roles in the KAIROS schema; and (2) arguments from the reasoning chains in Section 4.3 to create synthetic sentences describing the events. Subsequently, we modify these synthetic sentences by substituting the answer argument with an appropriate wh-word, facilitating the generation of synthetic sub-questions. Lastly, we generate the final multi-hop question by composing the sub-questions. The sub-questions naturally consist of the step-by-step explanations. The two blocks in the middle of Figure 2(b) demonstrate the process of generating synthetic question answer pairs & explanations from event reasoning chains.

### 4.5 Human Curations

These synthetic question answer pairs and explanations in the previous section may not be fluent and grammatically correct. Annotators are required to rephrase both of them in their own speaking style. They can edit anything or rewrite the whole question and all explanations. Moreover, since reasoning chains are sampled from all events in the documents of the dataset, automatically generated single answer (ending node of the reasoning chain) may not be complete. We ask annotators to find all answer spans after they finalize questions and explanations. Figure 2(b) includes the human curation process for turning synthetic questions into natural language questions. The word TRANSPORTED has been rephrased to TRAVELED, which sounds more natural. Details of crowd-sourcing and payment are shown in Appendix E.

## 5 Dataset Analysis

We collect 211 documents from WikiEvents [Li et al., 2021] to create our MEQA dataset and select five workers to annotate the dataset according to the five strategies illustrated in Table 1. After the annotation procedures, the dataset is further split into training, development, and test sets with a proportion of 80%:10%:10%. A summary of the general dataset statistics is shown in Table 7. We analyze our question type distributions and data quality in this section, and some additional analysis are shown in Appendix F.

### 5.1 Question Type Distribution

The distribution across the five strategies is organized in Table 1. From the table, it is evident that the first two strategies event-bridging and entity-bridging questions account for the largest percentage with a proportion of 49.7% and 37.1%, due to the rich event trigger and argument annotations in WikiEvents. For the rest of the question types, the number of comparison questions is smaller due to the requirement of numbers mentioned in event arguments affiliated with the same events.

### 5.2 Data Difficulty Evaluation

We design three prompting-based methods to test on datasets. Table 2 illustrates that our MEQA dataset is the most challenging compared to HotpotQA and 2WikiMultihop.

Table 2: Performance on different methods over HotpotQA, 2WikiMultihopQA, and MEQA.

| | Precision | Recall | F1 |
|---|---|---|---|
| **ChatGPT (GPT-3.5-turbo-1106)** | | | |
| HotpotQA | 0.745 | 0.779 | 0.733 |
| 2WikiMultihop | 0.501 | 0.724 | 0.532 |
| MEQA | 0.190 | 0.536 | 0.238 |
| **ChatGPT CoT-QA (+ Entity)** | | | |
| HotpotQA | 0.777 | 0.813 | 0.763 |
| 2WikiMultihop | 0.534 | 0.757 | 0.565 |
| MEQA | 0.364 | 0.394 | 0.350 |
| **ChatGPT CoT-QA (+ Event Triggers)** | | | |
| MEQA | 0.321 | 0.377 | 0.312 |
| **Human** | | | |
| MEQA | 0.783 | 0.836 | 0.811 |

Table 3: HotpotQA entity-centric example (top) and MEQA event-centric example (bottom).

| | |
|---|---|
| Paragraphs | Paragraph A: The 2015 Diamond Head Classic was a college basketball tournament . . . **Buddy Hield** *was named the tournament's MVP*. Paragraph B: **Chavano Rainier "Buddy" Hield** is a Bahamian professional basketball player for the ***Sacramento Kings*** of the NBA... |
| Question | Which team does the player named 2015 Diamond Head Classic's MVP play for? |
| Answer | Sacramento Kings |
| Document | [. . . ] nation's Defense Ministry confirmed that a ***major general*** was **killed** in Syria by an improvised explosive device, *Al-Monitor* online reported. [. . . ] In 2017, a ***lieutenant general*** was **killed** in the same province, [. . . ] |
| Question | Who **died** before *Al-Monitor* reported online? |
| Answer | major general, lieutenant general |

Employing ChatGPT as the foundational method, where each input contains only a context and a question and the output is only an answer (Appendix G), reveals that MEQA presents the greatest challenge. Additionally, our exploration of the effectiveness of the Chain-of-Thought [Wei et al., 2022] style prompt, denoted as "CoT-QA", in which a list of question answer pairs represents reasoning chains before answers. For "CoT-QA (+Entity)" and "CoT-QA (+Event Triggers)", extracted entities and lists of event triggers are leveraged in the context in the prompt correspondingly [Li and Du, 2023b]. Results are consistent across all three models.

We obtain human performance from 3 annotators (graduate students) on 100 samples randomly chosen from the test split. The accuracy is 88% on average, the upper bound of human performance is 92%, and human agreement [Cohen, 1960; Trivedi et al., 2022] is 79%.

Table 3 presents examples comparing between HotpotQA and MEQA. The upper example is from the entity-centric benchmark HotpotQA. Its reasoning type is "entity bridging". The reasoning starts from "MVP", hops by "Buddy Hield", and ends with the answer "Sacramento Kings". To answer the question in the bottom example, models should start reasoning from the Al-Monitor and first locate the reported event; then find all events that happened before the reported event; and finally extract victims in all those events, which are answers to the question.

# 6 Experimental Results

## 6.1 Evaluation Metrics

In all experiments, we evaluate both answers and explanations. We compare generated answers with golden answers and report precision, recall, and F1-score. Specifically, we follow the evaluation script from HotpotQA. To evaluate the hallucination problem of explanations, we introduce novel new metrics including "completeness" and "logical consistency" [Huang et al., 2023].

**Completeness** refers to the step-wise accuracy of the matches between the golden explanations and the predicted explanations. We report precision, recall, and F1-score. Predicted explanations can

have multiple formats, such as QA-format or freeform format. We can always split them into smaller sub-steps, such as one question-answer pair or sentence.

For each question, golden reasoning explanations consist of lists of QA pairs obtained via the method in Section 4.1. We design an algorithm to count the number of matches between predicted and golden explanations. The core idea is that we compare explanations step-wise and follow the order of the golden explanation steps. If one golden step is unmatched, it won't be matched later. Predicted explanations are single-hop questions in which sub-step may contain one or two golden steps. The pseudo-code and prompt for calculating the metric is illustrated in Appendix H.

**Logical Consistency** is a reference-free metric. It measures whether each sub-step is consistent with the previous steps or the original question [Golovneva et al., 2023; Huang et al., 2023]. If there is no logical contradiction for a sub-step, we count it as logically consistent. Otherwise, we count it as inconsistent. More specially, we input the history and the current step to LLM to determine whether they have any logical contradiction.

## 6.2 Experimental Settings

We benchmark various methods based on prompting and fine-tuning. In all experiments, we utilize the few-shot method to include demonstrations as input. In the following experiments, we denote context as **C**, question as **Q**, QA-format explanation as **E**, freeform explanation as **FE**, and answer as **A**. We use $\rightarrow$ to connect the instruction/input contents and output formats. All experiments using LLM are performed using ChatGPT (`GPT-3.5-turbo-1106`). Detailed prompts are in Appendix G.

We evaluate the performance of a variety of methods on our MEQA dataset. For prompting-based method, we include the following variations: (1) base fewshot (C+Q$\rightarrow$A); (2) CoT-QA (C+Q$\rightarrow$E+A); and (3) CoT-Freeform (C+Q$\rightarrow$FE+A). The "CoT-Freeform" refers to the traditional CoT method in Wei et al. [2022] in which freeform sentences compose the reasoning chains. We also design fine-tuning based methods using T5 [Raffel et al., 2023] as the base model.

In addition, we explore the impact of incorporating *golden* structured information (such as event graphs) on this event-centric multi-hop question answering task. This investigation is motivated by two key factors: (1) golden structured information represents the upper bound of additional information, and (2) the study conducted by Li et al. [2023] has demonstrated its effectiveness. More specifically, The "Entity" structure exclusively consists of lists of entities. In contrast, the "Entity KG" structure encompasses extracted entity-based knowledge graphs presented in a triple format derived from contexts. The "Event Triggers" structure solely encompasses triggers from corresponding events. Meanwhile, the "Event Triggers + Arguments" structure encompasses all trigger-argument pairs, along with trigger-trigger pairs if temporal or causality relations exist between two events. Finally, the "Full Event KG" structure represents a comprehensive iteration of the aforementioned structure. It incorporates roles between trigger-argument pairs and relations between events, thus providing a more exhaustive representation. We add the variations of the four baselines as different groups.

## 6.3 Experimental Results

The main results are in Table 4. More experiments using Claude and Llama3 are in Appendix I. As discussed in Li and Du [2023b], recall is a better metric for evaluating LLMs' performance. The "CoT-Freeform" method has the highest recall value among baselines of each group, but it does not significantly differ from the "Fewshot" method. The performance of "CoT-QA" is the worst because of the challenge of generating the sequence of QA pair-based explanations which lead to the final answer: (1) QA-based explanations contain fruitful intermediate information and are mandatory outputs; (2) they are harder to generate: all intermediate questions should have logical connections and be faithful to the context. We can also observe that the precision of the "CoT-QA" method is the best. One main reason is that the format of QA-based explanation successfully guided ChatGPT to output short answers. Moreover, the discussion of potential leakages of the dataset is in Appendix J.

Table 5 presents the performance of different question types in the Full Event KG setting using GPT-3.5. "Entity Bridging" achieves the highest overall performance, likely because it is the simplest question type and similar questions are commonly encountered by LLMs. In contrast, "Event Relation" has the lowest recall, as it involves more complex reasoning process.

Table 4: Performance on all experiments. Four baselines and their further experiments are grouped in the table. In each group, the first line is the performance of the baseline. All the following lines in a group indicate additional contents that are appended after context **C**. **Bold numbers** shows the best results in each column. Numbers with (*) indicate they are the best among all baselines.

| Method | General Performance | | | Completeness | | | Logical Consistency |
|---|---|---|---|---|---|---|---|
| | Precision | Recall | F1 | Precision | Recall | F1 | |
| T5 (C+Q→A) | 0.3012* | 0.2761 | 0.2831 | - | - | - | - |
| *w/ Entity KG* | 0.3187 | 0.2813 | 0.2942 | - | - | - | - |
| Fewshot (C+Q→A) | 0.1902 | 0.5360 | 0.2377 | - | - | - | - |
| *w/ Full Event KG* | 0.4541 | 0.6355 | 0.4581 | - | - | - | - |
| CoT-QA (C+Q→E+A) | 0.2832 | 0.3903 | 0.2940* | 0.1963 | 0.2141* | 0.2001 | 0.6442 |
| *w/ Entity* | 0.3636 | 0.3943 | 0.3500 | 0.2052 | 0.2321 | 0.2145 | 0.6161 |
| *w/ Entity KG* | 0.3522 | 0.3913 | 0.3344 | 0.1935 | 0.2118 | 0.1978 | 0.6318 |
| *w/ Event Triggers* | 0.3210 | 0.3773 | 0.3120 | 0.2792 | 0.2946 | 0.2835 | 0.6693 |
| *w/ Event Triggers + Arguments* | 0.4910 | 0.4878 | 0.4471 | 0.3431 | 0.3698 | 0.3481 | 0.6553 |
| *w/ Full Event KG* | **0.5299** | 0.5298 | **0.4940** | 0.3989 | **0.4653** | **0.4208** | 0.7327 |
| CoT-Freeform (C+Q→FE+A) | 0.1044 | 0.5392* | 0.1494 | 0.3368* | 0.1678 | 0.2161* | **0.9132*** |
| *w/ Full Event KG* | 0.3680 | **0.6575** | 0.3823 | **0.4566** | 0.2506 | 0.3145 | 0.8889 |

**Explanations from CoT-QA are more faithful than CoT-Freeform.** Comparing completeness metrics, "CoT-QA" shows higher recall but lower precision, suggesting it produces more explanations matching golden explanations but also generates redundant QA pairs, potentially leading to hallucination issues. In contrast, "CoT-Freeform" explanations, while shorter, are less related to golden explanations, indicating fidelity issues and potential hallucination. Notably, "CoT-QA w/ Full Event KG" outperforms all models in recall and F1 score, implying better alignment with golden explanations. Incorporating comprehensive structured data notably enhances both completeness and faithfulness.

Table 5: Performance of question types on GPT-3.5.

| GPT-3.5-turbo-1106 CoT (Full Event KG) | Performance | | |
|---|---|---|---|
| | Precision | Recall | F1 |
| Event Relation | 0.4740 | 0.4492 | 0.4265 |
| Entity Bridging | 0.5539 | 0.5404 | 0.5094 |
| Event Listing and Counting | 0.3895 | 0.5024 | 0.4049 |
| Event Comparison | 0.3682 | 0.4622 | 0.3963 |

Logical consistency results of the "CoT-QA" and "CoT-Freeform" are not comparable. The reason that "CoT-Freeform" has such a high consistency score is its length. Since ChatGPT generated a shorter reasoning chain without multi-hop explanation, our evaluation script always counts explanations as consistent if the results are correct. Also, even if the result was incorrect, generated explanations always repeated partial original questions. These explanations were always counted as consistent.

**Rich structured information improves performance.** We set several levels of structured information as additional information added in inputs. According to the richness of structured information, we rank additional structures as following: Entity ≈ Event Triggers < Entity KG < Event Triggers + Arguments < Full Event KG.

The performance trend in Table 4 correlates with the richness of additional event-related structured information. Models tend to perform better when they are provided with richer information. Across all groups (grey rows), the performance with "Full Event KG" is always better than others, not only because of its graph structure information but also its explicit relations between all elements. An example is shown in Figure 4, in which the explanations are not applicable unless the complete event KG is supplied. The context and further analysis of additional information is in Appendix K.

**Human Correlation Studies.** We sample 100 questions from "CoT-QA" (Full Event KG) and ask annotators to follow the definition of completeness and logical consistency to judge the quality of generated explanations. Annotators report the number of matched explanation steps, and then calculate the correlations between automatically generated scores and human evaluations.

The correlation of the "completeness" and "logical consistency" metrics are 0.693 and 0.601, respectively, Both correlations exceed 0.35, a threshold deemed moderate according to Taylor [1990]. They satisfy the requirement of the most recent LLM reasoning evaluation work by Huang et al. [2023]. Thus, our metrics are usable to evaluate explanations for this specific aspect.

Table 6: An example of comparing two CoT methods. Context is in Table 13.

**Question:** *Which object was mentioned in a discussion before given to Dzhokhar?*
**Answer:** *Pistol*

**CoT-QA Explanation:**
1. What event contains Dzhokhar as the recipient of an object? borrowing
2. What event before #1 involves the object? discuss
3. What was the object in #2? Pistol
**Answer:** *Pistol*

**CoT-Freeform Explanation:**
Before giving the object to Dzhokhar, the object was discussed among [...]
**Answer:** *Pistol*

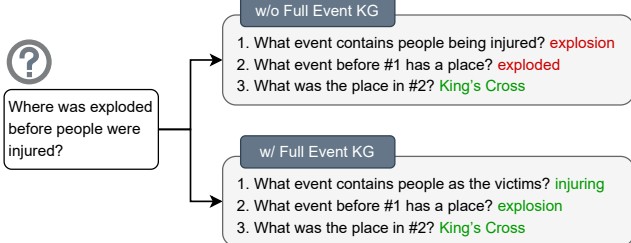

Figure 4: An example of the significance of additional structured information.

We also provide a comprehensive error analysis over randomly selected 50 samples. There are two major types of errors: *Incorrect Start Event Identification* and *Incorrect Event Relation Identification*. Details are in Appendix L.

## 6.4 Comparing CoT-QA and CoT-Freeform

In our experiment, we encounter scenarios where both CoT-QA and CoT-Freeform methods exhibit effective performance in the process of answer generation. As shown in Table 6, CoT-QA excels in providing detailed, specific answers by breaking down queries into smaller, structured components; however, it may not always align closely with the main focus of the question. CoT-Freeform offers a broader context and tends to address queries more holistically, but it can sometimes lack the specific details in explanations required by the question. These insights suggest that while both approaches have their unique strengths, a tailored application based on the specific needs of the query might yield the best results.

## 7 Conclusion, Limitation, and Societial Impacts

**Conclusion.** We introduce MEQA, the first benchmark containing multi-hop questions requiring reasoning about both entities and events, as well as explanations. It is annotated based on our new QA generation strategy — utilizing document-level event structures to bootstrap natural questions in an efficient way. We demonstrate the potential and quality of this new dataset through a detailed analysis of its contents. We conduct experiments and show that the benchmark is challenging for a variety of state-of-the-art models, especially that they are prone to incompleteness and inconsistency issues when generating the reasoning explanations.

**Limitation.** Until now, MEQA only covers English documents which limits the advancement of event-centric multi-hop QA in multilingual scenarios. In the future, we plan to extend MEQA to evaluate the performance of language models in more languages.

**Societial Impacts.** Our benchmark could facilitate the development of more intelligent multi-hop event-centric QA models, which has substantial impacts in a variety of applications, e.g., news understanding, emergency response, etc.

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

# A   Definition and Complexity of Event-centric Questions

An event-centric question is a query that focuses on events, typically involving **event triggers** and their **associated argument entities**, often in the context of event structures and their relations.

We adopt ACE (Automatic Content Extraction) [Walker et al., 2006] definition of entities, relations, and events. Under this definition, an event consists of an event trigger (e.g., "kills") and its associated argument entities (e.g., "major general", "Syria"). An event trigger is a word that most clearly describes the event's occurrence, and it is **often a verb that evokes the action or the status** of the target event. Event arguments are mostly entities and need to be inferred in reasoning processes. Suppose an event structure is ($\textbf{Tri}$, $Arg_1$, $Arg_2$, $Arg_3$, ...). The event-centric QA can involve two types of relations: (i) between event $E$ (triggered by $\textbf{Tri}$) and across its certain arguments ($Arg_1$, $Arg_2$, ...), e.g., "major general" is a VICTIM of "kills"; or (ii) between event $E_1$ (triggered by $\textbf{Tri}_1$) and event $E_2$ (triggered by $\textbf{Tri}_2$), e.g., "kills" is BEFORE "reported". Different from traditional multi-hop QA, for any two argument entities ($Arg_1$, $Arg_2$) in an event, there is no *direct* relationship (bypassing event trigger) between $Arg_1$ and $Arg_2$. For instance, in the event ($\textbf{Tri}$=kills, $Arg_1$=LOCATION, $Arg_2$=TIMESTAMP, ...), there exists no direct relation between the location and the timestamp.

Moreover, two specific differences exist:

**More Complex Relations.** Unlike entity-entity relations (in entity-centric datasets), which typically feature a singular type of relation connecting all entity pairs, our event-centric dataset encompasses the above relation and also includes event-event relations and event-argument relations, which are much more complex. Below, we present examples and explanations of various types of relations:

a. *Event-Event Temporal Relations*: Any pair of events ($E_1$, $E_2$) have inherent temporal relations, such as before, after, and contains [Han et al., 2021]. For example, in the example (**Reported**, *after*, **Kills**), where **Bold texts** denote event triggers while the *italics text* indicate temporal relations, the temporal relation "*after*" connects two events: **Reported** and **Kills**. All of them cannot be directly extracted from the given context. Instead, models have to identify and infer them as part of their reasoning process.

b. *Event-Event Causal Relations*: A pair of events ($E_1$, $E_2$) exhibits a casual relation if $E_2$ definitely happens after $E_1$ [Han et al., 2021]. For example, (**Kills**, *caused by*, **Explode**), where **Bold texts** denote events while the *italics text* represents a causality relation, comes from the reasoning process to answer the question, "*What caused the death of the 21 people?*". Such relationships are inherently complex. While we can say someone/something causes an event, we can hardly say someone/something causes an entity. Consequently, this type of relationship is rarely encountered in current multi-hop datasets.

c. *Event-Entity Relations*: (**Pay**, timestamp, 11 am) is an example connecting trigger "**Pay**" and entity "11 am" by the relation "timestamp". This only exists in event-centric datasets.

d. *Entity-Entity Relations*: In the example, (Alice, daughter of, Mary), the "daughter of" links "Alice" and "Mary". The majority of relations connecting entities can be readily identified in the given context, while only a handful need rephrasing for clarity. Entity-centric datasets only contain this type of relation.

**More Complex Reasoning Process.** Our event-centric dataset presents a greater challenge compared to other multi-hop entity-entity datasets. While solving multi-hop questions in conventional datasets involves hopping over entities, in our dataset, models must navigate through both entities and events (a heterogeneous reasoning chain). This means that reasoning begins with entities as usual, but then extends to identifying the event associated with the entity, its location, and the trigger word for events, which poses significant difficulty for current models. There are two examples of two types of reasoning:

a. Hopping over only entities looks like: Alice $\xrightarrow{\text{daughter of}}$ Mary $\xrightarrow{\text{spouse of}}$ William. The reasoning chain contains only entities and relations in parentheses. This answers the question: "*Who is Alice's father?*". Given the direct relations between all entities in the provided context, models can easily answer the question by reasoning over these entities.

b. Hopping over both entities and events looks like: Al-Monitor $\xrightarrow{\textit{communicator}}$ **Reported** $\xrightarrow{\textit{after}}$ **Kills** $\xrightarrow{\text{victim}}$ General. **Bold texts** denote triggers, while the *italics text* indicate temporal

relations. It answers the question: "*Who was the victim before Al-Monitor reported online?*".
Unlike the above case with direct entity relations, there's no direct relation between "Al-Monitor" and "General" in the context provided. Consequently, simple entity hopping won't suffice. The reasoning process of the question requires identifying the event associated with "Al-Monitor" (the "**Reported**" event), establishing its temporal relation with the "**Kills**" event, and deducing the answer, "General". The event and relation identification process poses a considerable challenge for current models.

## B    Examples of Strategies

The following paragraphs explain examples in Table 1. Figures 2(a) and 2(b) correspond to the strategies 1 and 2 in Table 1.

**Strategy 1.** In Figure 2(a), the demise of a major general was conveyed by AI-Monitor, where "reported" functions as the event trigger and "AI-Monitor" assumes the role of "communicator", establishing an event-argument relation. Additionally, "kills" also acts as an event trigger, preceding the "reported" event; hence, a temporal relationship termed "BEFORE" exists between the two events. Moreover, entity-only reasoning chains fall short of capturing the entire reasoning process, particularly in establishing the relationship between "Al-monitor" and "General". Relations are always brief, precise, and general. The relation "reported" can not precisely convey the relationship, and "reported before the death of" lacks generality. Thus, there is no relation between them.

**Strategy 2.** In Figure 2(b), all green nodes are entities, and all blue triangle nodes are triggers of events. Edges connecting them are event-argument relations. The reasoning chain starts with "11 people". and hops with the bridging entity "Belfast" to the result "Plane". The events "exploded" and "transported" are connected by bridging entities but not direct event-event relations. Moreover, relying solely on entity-based reasoning chains may inadequately capture the entirety of the reasoning process. For the first sub-question, the context indicates "11 people" died due to the explosion at "Belfast", without a direct relationship. Even any implicit relationship can only provide partial information, like (11 people, died at, Belfast) In the second sub-question, no relation exists between "Belfast" and "Plane". They can only be connected by the event trigger "transported".

**Strategy 3.** In the context shown in Table 1, there are five events that describe "victims", such as the "kills" event, the "wounded" event, the "killed" event, etc. Models need to identify those events first and conduct the counter later. Due to the space limitation, only three are shown in the context.

**Strategy 4.** The question asks to compare the number of victims caused by IED or IS. Firstly, they belong to different roles. IED is an item, while IS is an organization. Then, models should extract all related events and count the number of victims among events respectively. IED kills 4 victims, while IS kills 24 victims. Finally, by comparing 4 and 24, models should choose 24 and output "IS" as the answer.

**Strategy 5:** This question is unanswerable since the reasoning chain breaks at the start step. The reasoning chain starts with "21 bombs". However, the context describes "*21 bombs have gone off in Belfast*" but does not mention where they are manufactured. We cannot move the reasoning step forward anymore. Thus, the question is unanswerable.

## C    WikiEvents and New Documents

WikiEvents [Li et al., 2021] dataset contains real-world news articles and annotates all events with the KAIROS ontology. It provides 246 documents and 33 event types defined in two levels. The first level covers eight general domains: life, movement, transaction, business, conflict, contact, personnel, and justice.

In this paper, we generate our dataset from the WikiEvent dataset, which contains over 15 scenarios across all domains. **Plus, our event-centric question-generation process is versatile across domains and languages**. Specifically, we can first perform event extraction (EE) to construct the event structures for every new domain and language and generate QA pairs datasets. For instance, we perform a preliminary study on the TORQUE [Ning et al., 2020] dataset, which does not include predefined events and event structures.

Given a new document from TORQUE:

> *Mr. Erdogan accepted the Israeli apology, the prime minister's office said. Mr. Erdogan has long sought an apology for the raid in May 2010 on the Mavi Marmara, which was part of a flotilla that sought to break Israel's blockade of Gaza.*

the first step is to perform event extraction by LLM. Event samples are as follows:

```
Event₁:
Trigger₁:  accepts
  Agent:  Erdogan
  Theme:  Israeli apology
Event₂:
Trigger₂:  raid
  Target:  Mavi Marmara
  Date:  May 2010
...
```

By following the question generation step we propose, we can get the following question:

> *What target in an old event caused the Israelis to apologize?*

We annotate 10 questions and ask two annotators to answer them. The average accuracy is 85%, and the upper bound accuracy is 90%. The human agreement (Cohen Kappa) score is 62%.

## D   KAIROS Schema Templates

For example, the template describing a transport event looks like:

> [**Transporter**] transported in [**Vehicle**] from [**Origin**] place to [**Destination**] place.

Tokens in square brackets are placeholders. Moreover, we split the above template into several chunks by its context and semantic meaning, such as:

- [**Transporter**]
- transported
- in [**Vehicle**]
- from [**Origin**] place
- to [**Destination**] place

In this format, we can keep useful chunks only and remove redundant parts to form new templates. Specifically, verbs are always kept in the final template. Each template can also be converted to a question-version template. In detail, the target placeholder in the template will be replaced by wh-words according to its corresponding semantics.

## E   Crowdsourcing Hiring and Payment

We hire student workers to annotate the dataset. All annotators have bachelor's degrees and basic knowledge about events, including trigger, role, and argument. Before annotation, we post a qualification test for all candidates, which includes a document and five template-based questions. We also provide an instruction document for all candidates to learn how to annotate questions. In this process, we answer all their questions about the annotation process to ensure they fully understand the task. Then, They annotate the template document. We select the top five workers who annotate accurately and quickly. In the annotation process, our workers were encouraged to spend **2-3** minutes per question on average. We paid around $0.65 per question. Among all workers, the average hourly rate is $15.

Table 7: General statistics of MEQA.

| | Train | Dev | Test | Overall |
|---|---|---|---|---|
| # of Documents | 189 | 20 | 22 | 211 |
| # of Questions | 1,674 | 282 | 287 | 2,243 |
| # of Events | 1,361 | 129 | 204 | 1,565 |
| # of Entities | 2,280 | 233 | 364 | 2,644 |
| Avg. Document Length | 695.4 | 815.2 | 494.9 | 676.4 |
| Avg. Question Length | 11.5 | 11.6 | 11.7 | 11.6 |
| Avg. # of Steps/Hops | 2.6 | 2.6 | 2.6 | 2.6 |

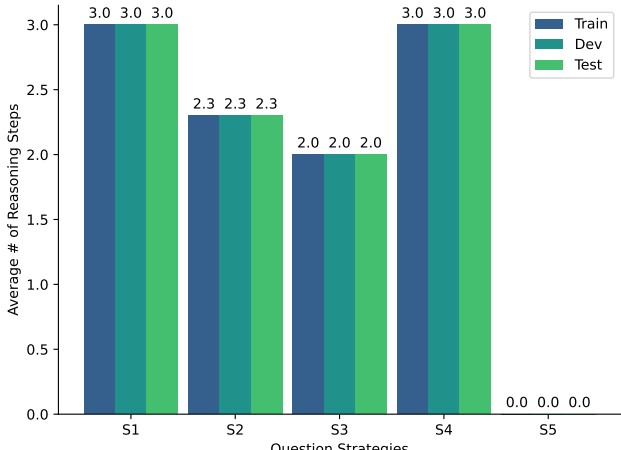

Figure 5: Average number of reasoning steps for each strategy in different data splits.

Table 8: Data statistics comparison between MEQA and existing datasets.

| | MEQA | HotpotQA | 2WikiMultihopQA | Musique |
|---|---|---|---|---|
| Avg. Question Length | 11.6 | 16 | 12.64 | 15.40 (tokens per question) |
| Avg. # of Hops | 2.6 | 2 | 2 | 2.4 |
| # of Documents | 211 | 14,810 (golden) | 25,152 (golden) | 7676 |
| # of Questions | 2,243 | 7,405 | 12,576 | 2,459 |
| # of Events | 1565 | - | - | - |
| # of Entities | 2644 | - | - | - |

# F   Additional Dataset Analysis

The general statistics of our MEQA dataset is shown in Table 7. We also analyze the number of reasoning steps and the most frequent $n$-grams in this section.

**Reasoning Steps.** Figure 5 contains the average reasoning steps of each strategy in different data splits, in which the distributions are almost the same in the different data splits, indicating the correctness of our dataset-splitting technique. Questions in Strategies 1 and 4 contain the most reasoning steps because the requirement of reasoning across the events and their arguments, and the steps of Strategies 2 and 3 are not comparable because two-step reasoning satisfies most questions. Finally, because the questions in Strategy 5 are unanswerable, we did not annotate the explanations for its questions.

**Most Frequent $n$-grams.** Figure 6 illustrates the most frequent $n$-grams, including uni-grams, bigrams, and trigrams in different strategies of questions, which can be regarded as semantic cues in the questions to reason about particular types of questions. From Fig. 6, we observe that the most frequent n-grams are consistent with the question types; for example, the most frequent $n$-gram in the event relation strategy includes specific event relations, such as "*before*" and "*because*", while those in the event listing and counting strategy include "*how many*" and "*related to*", indicating the consistency of the annotated questions with our proposed question strategies.

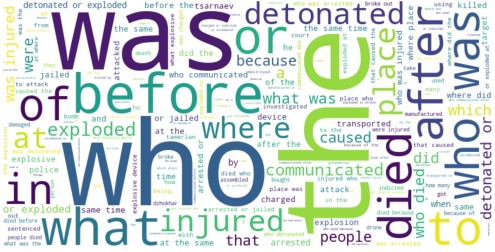

(a) The most frequent $n$-grams in Strategy 1.

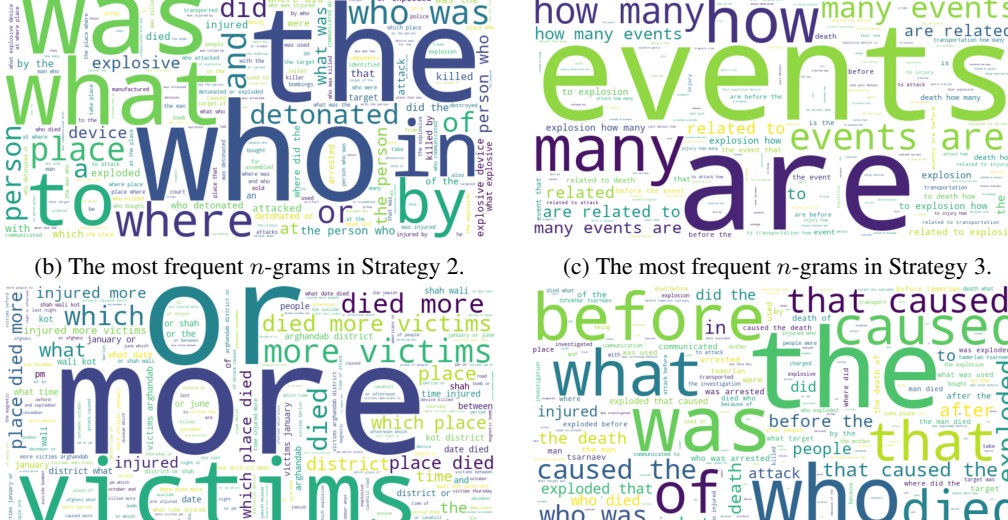

(b) The most frequent $n$-grams in Strategy 2.

(c) The most frequent $n$-grams in Strategy 3.

(d) The most frequent $n$-grams in Strategy 4.

(e) The most frequent $n$-grams in Strategy 5.

Figure 6: The most frequent $n$-grams in the questions in different strategies.

**Dataset Statistics Comparison.** Table 8 compares our MEQA with other datasets. Since those benchmark papers do not provide parts of the statistics, we estimate partial numbers based on the statistics they provide in the paper. Since none focus on events, they do not have event-related statistics.

## G  Prompts

Given the document and question, the entire prompt for a demonstration looks as follows:

**Vanilla Prompt.** This is the vanilla prompt for most question answering tasks. The input contains only documents and a question, while the output is the answer.

```
Document:
<Document>
Question:
<Question>
Answer:
<Answer>
```

**Chain-of-Thought (CoT).** The CoT prompt has reasoning chains before giving the final answer, while the Vanilla prompt does not contain it. In our implementation, reasoning chains are in question

answering format, which is more clear and easy for people to understand.

```
Document:
<Document>
Please answer the following question step-by-step with explanation:
<Question>
Answer:
<Question 1> <Answer 1>
...
<Question n> <Answer n>
So, the answer is:  <Answer>
```

**Post-hoc.** The post-hoc prompt provides explanations after generating the answer. The explanation format is the same as the format of reasoning chains in the CoT prompt.

```
Document:
<Document>
Please answer the following
question and then explain
reasoning steps:
<Question>
Answer:
<Answer>
Explanation:
<Question 1> <Answer 1>
...
<Question n> <Answer n>
```

**Event Relations.** Given the document and events, the prompt to generate relations between events is as follows:

```
Document:
<Document>
Event 1:
<Trigger1>
  <role1_1> <argument1_1>
  <role1_2> <argument1_2>
  ...
Event 2:
<Trigger2>
  <role2_1> <argument2_1>
  ...
Relation Options:
Before, After, At the same time, Contain, Contained by, Cause, Caused by, ...
Please output the relation between the two events in the above document:
```

## H   Pseudocode and Prompt of the Completeness Metric

The pseudocode of the algorithm is illustrated in Algorithm 1, whose time complexity is in quadratic time complexity $O(n^2)$. In this algorithm, a prediction step ($pi$) moves forward only if it cannot find any more matches to a golden step. A golden step moves forward only if it is matched. The "match" condition is the core of the algorithm.

**Algorithm 1** Incompleteness Algorithm

```
1:  pe {Predicted Explanation}
2:  ge {Golden Explanation}
3:  pi = 0, gi = 0, match = 0
4:  while pi < len(pe) do
5:      matchFlag = False
6:      for i in [gi, len(ge) − 1] do
7:          {Core condition}
8:          if pe[pi] match ge[i] then
9:              match+ = 1
10:             gi = i + 1
11:             matchFlag = True
12:             break
13:         end if
14:     end for
15:     if matchFlag == False then
16:         pi+ = 1
17:     end if
18: end while
```

There are two methods for the "match": string and semantic matching. In the string matching method, a prediction step matches a golden step if two elements in the golden triple can be found in the prediction step. It is straightforward and simple. However, texts are flexible, and a difference of even one character can cause a mismatch. For example, the golden triple is (IED, kills, two dozen fighters) and an explanation is "*Who was killed by IED? 24 fighters?*". The string matching returns "False" since only the IED is matched.

To solve the issue, we propose semantic matching by LLMs. It compares two question answering pairs instead of comparing a question answering pair with triples. The prompt looks as follows:

```
Please determine whether the following questions and answers
share the same semantic meanings respectively.
<Question 1> <Answer 1>
<Question 2> <Answer 2>
Please only output "yes" or "no":
```

We output "match" if the output is "yes". Otherwise, the two question-answer pairs mismatch.

Overall, semantic matching is recommended as the matching condition to achieve the best performance.

# I More Experimental Results

Tables 9 and 10 present the performance on "CoT-QA" and "CoT-Freeform" using Claude and Llama3 models. The precision scores of Claude-3 are low mainly because the model failed to follow the answer format and output redundant information in the outputs. In both tables, the trends of all metrics are the same as in Table 1 in the paper. Among GPT-3.5, Claude-3-Haiku, and Llama3-70B models, GPT-3.5 performs best considering all metrics.

# J Potential Data Leakage

This work mainly tests our dataset on GPT-3.5-turbo-1106 and T5 models. GPT-3.5-turbo-1106 was trained on data collected before Sep 2021[2]. T5 models were released in 2019. However, WikiEvents was released in Oct 2021. Thus, both models should have yet to see the WikiEvent data.

Moreover, even with potential leakages of the original WikiEvent dataset, since the new dataset focuses on reasoning, models do not remember reasoning processes. We design a new experiment

---

[2]https://platform.openai.com/docs/models/gpt-3-5-turbo

Table 9: Performance on all experiments using Claude-3-Haiku. Two baselines and their further experiments are grouped in the table. In each group, the first line is the performance of the baseline. All the following lines in a group indicate additional contents that are appended after context **C**. **Bold numbers** shows the best results in each column. Numbers with (*) indicate they are the best among all baselines.

| Method | General Performance | | | Completeness | | | Logical Consistency |
|---|---|---|---|---|---|---|---|
| | Precision | Recall | F1 | Precision | Recall | F1 | |
| CoT-QA (C+Q→E+A) | 0.0177 | 0.5460 | 0.0330 | 0.2639 | 0.3476* | 0.2403 | 0.6261 |
| *w/ Entity* | 0.0150 | 0.4860 | 0.0282 | 0.3399 | 0.3119 | 0.2578 | 0.6529 |
| *w/ Entity KG* | 0.0184 | 0.5395 | 0.0346 | 0.3552 | 0.3643 | 0.2881 | 0.5911 |
| *w/ Event Triggers* | 0.0216 | 0.4979 | 0.0396 | 0.2908 | 0.3857 | 0.2722 | 0.6783 |
| *w/ Event Triggers + Arguments* | 0.0250 | 0.5832 | 0.0466 | 0.3782 | 0.3905 | 0.3109 | 0.6488 |
| *w/ Full Event KG* | **0.0245** | 0.5929 | 0.0454 | 0.3480 | **0.3976** | 0.3045 | 0.6834 |
| CoT-Freeform (C+Q→FE+A) | 0.0207* | 0.6040* | 0.0395* | 0.3714* | 0.2595 | 0.3043* | 0.8929* |
| *w/ Full Event KG* | 0.0251 | **0.6430** | **0.0475** | **0.5286** | 0.3714 | **0.4343** | **0.9071** |

Table 10: Performance on all experiments using Llama3-70B.

| Method | General Performance | | | Completeness | | | Logical Consistency |
|---|---|---|---|---|---|---|---|
| | Precision | Recall | F1 | Precision | Recall | F1 | |
| CoT-QA (C+Q→E+A) | 0.2056* | 0.5311 | 0.2268* | 0.1542 | 0.2190* | 0.1790 | 0.5124 |
| *w/ Entity* | 0.2014 | 0.5368 | 0.2488 | 0.1933 | 0.2119 | 0.1906 | 0.5212 |
| *w/ Entity KG* | 0.2423 | 0.5434 | 0.2812 | 0.1910 | 0.2548 | 0.2164 | 0.5415 |
| *w/ Event Triggers* | 0.2526 | 0.5533 | 0.2973 | 0.1951 | 0.2810 | 0.2289 | 0.5437 |
| *w/ Event Triggers + Arguments* | 0.2691 | 0.5474 | 0.2988 | 0.2788 | 0.3833 | 0.3204 | 0.5311 |
| *w/ Full Event KG* | 0.3021 | **0.5950** | **0.3277** | 0.3429 | **0.4274** | **0.3793** | 0.5599 |
| CoT-Freeform (C+Q→FE+A) | 0.1363 | 0.5356* | 0.1740 | 0.2643* | 0.1786 | 0.2114* | **0.9129*** |
| *w/ Full Event KG* | **0.3120** | 0.5445 | 0.3253 | **0.3929** | 0.2738 | 0.3214 | 0.9071 |

to compare the performance of two models. In this experiment, we apply two models, GPT-4-0613 (training data up to Sep 2021 before WikiEvent) and GPT-4-turbo-2024-04-09 (training data up to Dec 2023 after WikiEvent). The two models have similar abilities across other NLP tasks, such as math problems (on GSM8k and MATH), open-domain question answering (on Natual Question, ARC), text summarization, etc, but the GPT-4-turbo-2024-04-09 may be affected by the potential leakage.

Based on Table 11, the performance of GPT-4-turbo-2024-04-09 does not exceed its of GPT-4-0613. Thus, the potential leakage of WikeEvent has little influence over our MEQA benchmark evaluation on event-based reasoning capabilities. As a similar finding is in Trivedi et al. [2022], models pre-trained on Wikipedia do not perform well on the multi-hop questions requiring complex reasoning.

## K  Additional Information Analysis

The context of the example is shown in Table 12. Since the prompts for QA-form and Free-form explanations are different, the results of "CoT-QA" and "CoT-Freeform" settings are not directly comparable. Using discrete entities or triggers as additional information (Entity or Event Triggers) can hardly affect the performance of ChatGPT, and sometimes, they can mislead the model because of their simplicity and incompleteness, leading the model to hardly learn helpful information from the graphs. However, after combining the entities and trigger information (Event Triggers + Arguments), more information was composed in the event structure format, which significantly helped the model find event-related information and answer event-centric questions.

## L  Error Analysis

**Incorrect Start Event Identification.** Because of the sequential nature of natural language, the model tended to read and reason questions from the beginning, which isn't always the case in real reasoning scenarios. For example, given the question "*What was damaged before noticed by individuals?*",

Table 11: Results of GPT-4-0613 and GPT-4-turbo-2024-04-09.

| | Performance | | | Completeness | | | Consistency |
|---|---|---|---|---|---|---|---|
| | Precision | Recall | F1 | Precision | Recall | F1 | |
| **GPT-4-0613** | | | | | | | |
| CoT-QA (C+Q→E+A) | 0.0674 | 0.5269 | 0.1049 | 0.2155 | 0.1857 | 0.1941 | 0.6298 |
| *w/ Full Event KG* | 0.2429 | 0.5537 | 0.2615 | 0.4500 | 0.4971 | 0.4664 | 0.6474 |
| CoT-Freeform (C+Q→FE+A) | 0.2151 | 0.5310 | 0.2598 | 0.4786 | 0.2214 | 0.2915 | 0.9286 |
| *w/ Full Event KG* | 0.5718 | 0.5412 | 0.5310 | 0.5071 | 0.2833 | 0.3529 | 0.9573 |
| **GPT-4-turbo-2024-04-09** | | | | | | | |
| CoT-QA (C+Q→E+A) | 0.0400 | 0.5418 | 0.0573 | 0.1945 | 0.2167 | 0.2018 | 0.6210 |
| *w/ Full Event KG* | 0.1057 | 0.5586 | 0.1191 | 0.4331 | 0.4810 | 0.4523 | 0.6540 |
| CoT-Freeform (C+Q→FE+A) | 0.2678 | 0.4918 | 0.3056 | 0.3643 | 0.2457 | 0.3071 | 0.9373 |
| *w/ Full Event KG* | 0.5467 | 0.5384 | 0.5206 | 0.4143 | 0.2881 | 0.3386 | 0.9455 |

Table 12: Context for the further experiments example in Figure 4.

**Context:**

1973: Bomb blasts rock central London. Scotland Yard is hunting a teenage suspect after two bombs at mainline stations injured 13 people and brought chaos to central London. The first explosion at King's Cross - which injured five people - occurred seconds after a witness saw a youth throw a bag into a booking hall. Fifty minutes later a second blast rocked a snack bar at Euston station, injuring a further eight people. No group has yet said it planted the bombs, but police have said the 2-3 lb (0.9-1.4 kg) bombs were typical of IRA manufacture. People were being thrown through the air King's Cross witness.

the model started its reasoning from "**damaged**" but not "**noticed**", thus it was unable to derive the correct answer based on the missing information.

**Incorrect Event Relation Identification.** Another major problem is that, even though the event relations were explicitly stated in the questions, models cannot memorize them, which may further invalidate the entire chain of reasoning. For example, although we explicitly mentioned the CAUSED BY relationship in the question "*Who was injured due to the explosion at Belfast?*", the model still returns an explanation with the AT THE SAME TIME relationship and makes an incorrect answer.

# M   Context for the Example Comparing CoT-QA and CoT-Freeform

Table 13: Context for the further experiments example in Table 6.

**Context:**
Here is a look at some of the critical pieces of evidence presented during the trial. Taken as a whole, the evidence suggests that the plan to bomb the Boston Marathon took shape over three months. A key issue for jurors — both in the guilt phase and later in the penalty phase if Tsarnaev is convicted — will be whether the jurors see Tsarnaev as an equal partner with his older brother, Tamerlan Tsarnaev, in the Boston Marathon bombing and the violent events that followed. As of Tuesday morning, jurors began reviewing evidence and witness testimony, which will play a role in helping them decide Dzhokhar Tsarnaev's guilt on each of the 30 charges he faces. January 2013 Tsarnaev, then 19, talks to his close friend, Stephen Silva, about borrowing Silva's 9mm P95 Ruger semi-automatic pistol. Tsarnaev and Silva both graduated from Cambridge Rindge and Latin School in 2001, and worked as lifeguards together at the Harvard pool. Starting in college, both became very involved in selling marijuana, and Silva had obtained a gun, in part, to help him protect his drug business. A Ruger semi-automatic handgun presented in the trial. US Attorney's " Office/REUTERS Jan. 30, 2013 Tamerlan, 26, allegedly buys two Fager pressure-cookers at Macy's at the Square One Mall in Saugus.The remains of a pressure cooker bomb. Josh Reynolds for The Boston Globe/Globe Freelance Feb. 8, 2013 Tamerlan Tsarnaev used his credit card to make an online purchase of a remote-controlled car set, batteries, and a transmitter and receiver from NitroRCX.com. Prosecutors said these items were used to help remotely-detonate the bombs February, 2013 Dzhokhar Tsarnaev visits Silva and borrows the Ruger pistol — the gun that was later used to kill MIT police officer Sean Collier and during the shootout with police in Watertown. [...]

# N   Annotation Interface

We illustrate our annotation interface for different strategies in Figures 7, 8, and 9, respectively. Figures 7 and 8 are the interfaces for Strategies 1 and 2, in which the KAIROS template-based questions are provided to annotators, who are required to annotate appropriate event relations for Strategy 1 and translate the template-based questions to natural languages. Regarding the annotations for Strategies 3, 4, and 5, as shown in Figures 9, annotators are not provided with template-based questions and are required to annotate the event listing and counting, event comparison, and unanswerable questions based on the event mentions and their annotations to the questions for Strategies 1 and 2. The full annotation guideline is shown in the GitHub repository.

**Annotation Instructions** (Click to collapse)

{{INSTRUCTIONS}} {{LINK}}

Enable developer mode: ☐

### Document

An unknown virus may be spreading within Venezuela's Aragua state, medical officials report; however, the state's governor denies these allegations and accuses the official of lying. According to the Venezuelan newspaper El Nacional, the crisis arose in Central Hospital of Maracay where personnel were seen with mouth covers. Social media began rumors about various deaths within the hospital caused by bacteria similar to Neisseria meningitides. Angel Sarmiento, president of Aragua Medical School, announced eight people died of the yet unidentified disease. He said it was not Ebola, meningitis, dengue or chikungunya. He added that the victims were four children and four adults. The symptoms exhibited by the dead were spots on the body that turned to boils, high fever, massive hemorrhaging stemming from smaller hemorrhagic incidents, and multi-organ failure. The disease took around 72 hours to run its course. "We don't know what we are confronting," Sarmiento said. "We don't know if it's a virus or bacteria. How can we heal what we don't know?" However, there may not be any disease to battle. The governor of Aragua, who is also the state's leader of the United Socialist Party of Venezuela, denies Sarmiento's claims, according to the Latin American Herald Tribune. Gov. Tarek el Aissami said, "I need to start by categorically denying the existence of some virus or bacteria on the premises of ▇▇▇▇▇▇▇▇▇ that is putting the lives of ▇▇▇ at risk." Aragua's governor refuted Sarmiento's claim and called them a "terrorist matrix that has a basic purpose, which is to create alarm, anguish in Aragua's population." He added that the deceased died of different causes, including an old man dying of diabetes and one of the children from leukemia. "What you are is a criminal, who, irresponsibly using your status as a doctor, launched this campaign without finding out the facts, without investigating, and in these last few hours you've caused terrible anguish among the people of Aragua," el Aissami said about Sarmiento. The governor called on the Attorney General's office to investigate Sarmiento.

You have to finish all the events before submitting. (Remember that you can't refresh the page otherwise the progress will be gone, to prevent this from happening, we suggest that you write the QA pairs in the google doc and copy paste them here)

[virus @ 2] [bacteria @ 70] [disease @ 93] [virus @ 271] [bacteria @ 273]

[broke out at where place when broke out among patients victims or population ▾]

*KAIROS event arguments for triggers*

| virus @ 2 | bacteria @ 70 |
|---|---|
| Disaster.DiseaseOutbreak.Unspecified | Disaster.DiseaseOutbreak.Unspecified |
| [Disease] disease broke out among [Victim] victims or population at [Place] place | [Disease] disease broke out among [Victim] victims or population at [Place] place |
| **Place:**state | **Place:**hospital |

| virus @ 271 | bacteria @ 273 |
|---|---|
| Disaster.DiseaseOutbreak.Unspecified | Disaster.DiseaseOutbreak.Unspecified |
| [Disease] disease broke out among [Victim] victims or population at [Place] place | [Disease] disease broke out among [Victim] victims or population at [Place] place |
| **Victim:**patients | **Victim:**patients |
| **Place:**the Maracay Central Hospital | **Place:**the Maracay Central Hospital |

*These are relations between events.*

**virus @ 2** [none ▾] **virus @ 271**

**virus @ 2** [none ▾] **bacteria @ 273**

**bacteria @ 70** [contains ▾] **virus @ 271**

**bacteria @ 70** [none ▾] **bacteria @ 273**

[ + ] Add a QA pair

[ − ] Remove

**question:** broke out at where place when broke out among patients victims or population    **answer:** state,hospital,state

**explanation:**  **step 1** What event contains patients is the Victim? bacteria

**step 2** What event is overlaps #1 has a Place? virus

**step 3** broke out at where place in the #2? state

[Submit]

Figure 7: An example annotation interface for annotating the questions for Strategy 1: Event Relations.

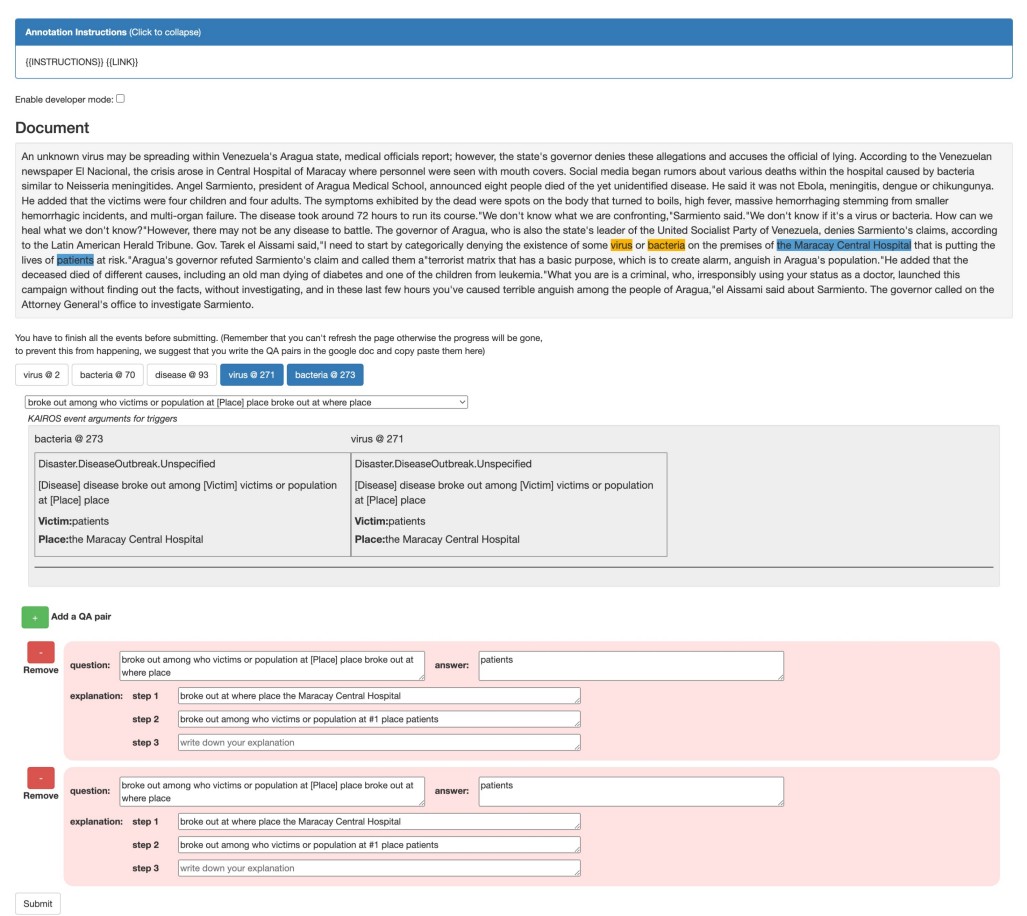

Figure 8: An example annotation interface for annotating the questions for Strategy 2: Entity Bridging.

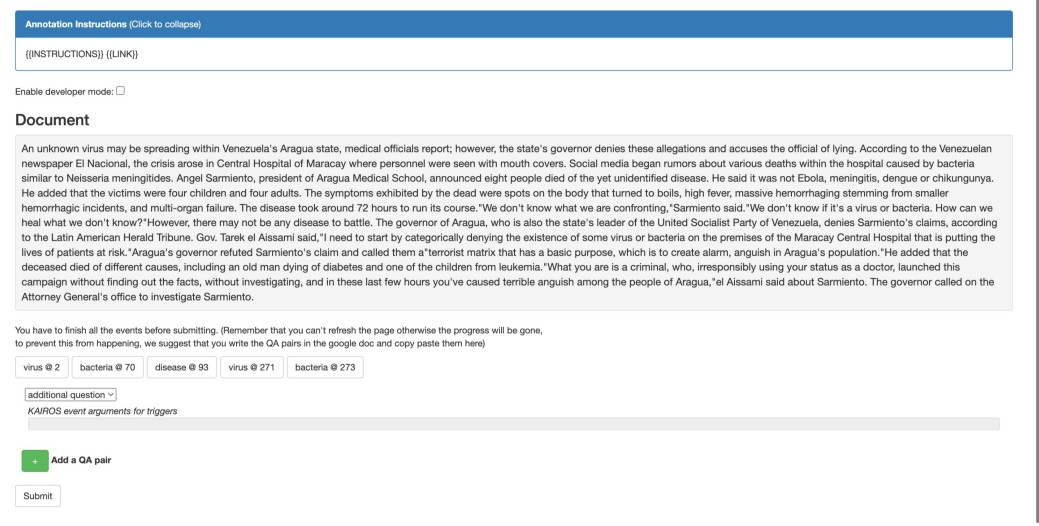

Figure 9: An example annotation interface for annotating the questions for Strategies 3 to 5: Event Listing and Counting, Event Comparison, and Unanswerable.

