# Supplementary Materials for
# MEQA: A Benchmark for Multi-hop Event-centric Question Answering with Explanations

**Ruosen Li, Zimu Wang, Son Quoc Tran, Lei Xia, Xinya Du**

Department of Computer Science, University of Texas at Dallas
{ruosen.li, zimu.wang, lei.xia, xinya.du}@utdallas.edu

## Contents

# 1 Data Format

We utilize an open and widely used data format, i.e., JSON format, for the MEQA dataset. A sample within the dataset, accompanied by the data format explanation, is shown in Listing 1.

```json
{
  "example_id": "dev_0_s1_3",  # An unique string for each question
  "context": "Roadside IED kills Russian major general [...]",  # The
    context of the question
  "question": "Who died before AI-monitor reported it online?", #  A
    multi-hop event-centric question
  "answer": "major general,local commander,lieutenant general", # The
    answer for the question
  "explanation": [
    "What event contains Al-Monitor is the communicator? reported",
    "What event is after #1 has a victim? killed",
    "Who died in the #2? major general,local commander,lieutenant
    general"
  ]  # A list of strings indicating the explanations (reasoning chain)
}
```

Listing 1: Data format for the MEQA dataset.

# 2 Data Availability, License, and Maintenance Plan

The dataset and source code for the MEQA dataset have been released to GitHub: `https://github.com/du-nlp-lab/MEQA`. We share the dataset under the MIT License (see `https://choosealicense.com/licenses/mit/`). We will actively maintain the dataset and make necessary updates to ensure its availability and quality.

# 3 Datasheets for the MEQA Dataset

We present a list of Datasheets [Gebru et al., 2021] for the MEQA dataset, synthesizing many of the other analyses we performed in this paper. Detailed data analysis can be found in Appendix F in the paper.

## 3.1 Motivation

1. For what purpose was the dataset created? *The dataset is created for multi-hop event-centric question answering (QA), a task much more challenging than entity-centric questions but overlooked by popular QA benchmarks.*

2. Who created the dataset and on behalf of which entity? *The dataset is created by Dr. Xinya Du's research group on behalf of the University of Texas at Dallas.*

3. Who funded the creation of the dataset? *The dataset is funded by Dr. Xinya Du. Appendix E in the paper describes payment details.*

## 3.2 Composition

1. What do the instances that comprise the dataset represent? *The instances that comprise the dataset include the documents, events, entities, event-centric multi-hop QA questions, answers, and explanations.*

2. How many instances are there in total? *The dataset includes 211 documents and 2,243 questions. Each question corresponds to an answer and a chain of explanations (reasoning chain). Details are in Section 5 of the paper.*

3. Does the dataset contain all possible instances or is it a sample of instances from a larger set? *The dataset contains all possible instances.*

4. What data does each instance consist of? *Each instance consists of a document, a multi-hop event-centric question, an answer, and a chain of explanation (reasoning chain).*

5. Is there a label or target associated with each instance? *Yes (the answer and explanations).*

6. Is any information missing from individual instances? *No.*

7. Are relationships between individual instances made explicit? *No.*

8. Are there recommended data splits? *Yes, we provide a recommended data split for training (1,674 questions), development (282 questions), and test sets (287 questions).*

9. Are there any errors, sources of noise, or redundancies in the dataset? *Because the dataset is human-annotated, it cannot be entirely correct. However, our MEQA dataset achieves a high quality, whose human performance is 88% on average, 92% on the upper bound, and the inter-annotator agreement is 79%.*

10. Is the dataset self-contained, or does it link to or otherwise rely on external resources? *The dataset is self-contained.*

11. Does the dataset contain data that might be considered confidential? *No.*

12. Does the dataset contain data that, if viewed directly, might be offensive, insulting, threatening, or might otherwise cause anxiety? *No.*

## 3.3 Collection Process

1. How was the data associated with each instance acquired? *The data associated with each instance was acquired from the WikiEvents dataset [Li et al., 2021].*

2. What mechanisms or procedures were used to collect the data? *We collected the documents, events, and entities from the WikiEvents dataset [Li et al., 2021]. The annotation of questions, answers, and reasoning chains followed the procedures in Section 4 of the main paper by both ChatGPT and human annotators.*

3. Who was involved in the data collection process and how were they compensated? *We hired 5 graduate student workers to annotate the dataset. Among all workers, the average hourly rate was $15.*

4. Over what timeframe was the data collected? *The dataset was created in the end of 2023.*

5. Were any ethical review processes conducted? *No.*

## 3.4 Preprocessing/Cleaning/Labeling

- Was any preprocessing/cleaning/labeling of the data done? *No.*

- Was the "raw" data saved in addition to the preprocessed/cleaned/labeled data? *We include the "raw" data (the documents) and the labeled data in the dataset.*

- Is the software that was used to preprocess/clean/label the data available? *Not applicable.*

## 3.5 Distribution

1. Will the dataset be distributed to third parties outside of the entity on behalf of which the dataset was created? *Yes.*

2. When will the dataset be distributed? *The dataset has been released on GitHub:* `https://github.com/du-nlp-lab/MEQA`.

3. Will the dataset be distributed under a copyright or other intellectual property (IP) license, and/or under applicable terms of use (ToU)? *The dataset is distrubuted under the MIT license.*

4. Have any third parties imposed IP-based or other restrictions on the data associated with the instances? *No.*

5. Do any export controls or other regulatory restrictions apply to the dataset or to individual instances? *No.*

### 3.6 Maintenance

1. Who will be supporting/hosting/maintaining the dataset? *The authors of the MEQA dataset will be responsible for its maintenance.*

2. How can the owner/curator/manager of the dataset be contacted? *The owners of the dataset can be contacted via email or posting an issue on GitHub.*

3. Is there an erratum? *No.*

4. Will the dataset be updated? *Yes, the dataset will be updated if necessary (e.g., correcting errors).*

5. Will older versions of the dataset continue to be supported/hosted/maintained? *Yes, all older versions of the dataset continue to be hosted for applicable comparative studies.*

6. If others want to extend/augment/build on/contribute to the dataset, is there a mechanism for them to do so? *Yes, they can contact via email.*