# OpenReview forum: "MEQA: A Benchmark for Multi-hop Event-centric Question Answering with Explanations"
_NeurIPS.cc/2024/Datasets_and_Benchmarks_Track — NeurIPS 2024 Track Datasets and Benchmarks Poster_

### Official Review · Reviewer_spJ2 · 2024-07-26
**MEQA review**

**Rating:** 5
**Confidence:** 4

**Review:**

Generally, I think this paper introduces a new perspective of utilizing IE datasets and an interesting multi-hop event-centric reasoning setting. However, I believe these are non-negligible limitations considering the proposed benchmark and the experiments.

**Strengths:**

1. This paper introduces a novel multi-stop event-centric reasoning task setting, which could provide insightful perspectives and findings in assessing reasoning abilities of models.
2. This paper designs a semi-automated data construction process from the IE dataset. This utilization of IE data offers a new direction for the IE community. Moreover, the proposed data construction method can be easily transferred to more datasets and scenarios.
3. The presentation of this paper is clear and easy to follow. The figures and tables in the paper clearly present the 5 different question types.

**Additional Feedback:**

N/A

**Clarity:**

The presentation of this paper is clear and easy to follow. The figures and tables in the paper clearly present the 5 different question types.

**Correctness:**

The claims in the submission is generally correct. The data construction process and experiments are appropriate. However, there needs more explanations for the new challenges of the proposed benchmark.

**Documentation:**

Yes

**Limitations:**

See Opportunities For Improvement

**Opportunities For Improvement:**

1. New Challenges: Although the authors claim that event-centric multi-hop reasoning is a novel task that presents novel challenges, they do not elaborate in detail. Compared to previous multi-hop reasoning datasets, what specific new challenges does MEQA introduce? Especially if we treat events as ``entities’’.
2. Question distribution: The distribution of question types is too long-tailed, particularly with event comparison questions accounting for only 0.6%, which translates to roughly 12 questions, which is insufficient for effectively evaluating models. Additionally, it would be better to provide the performance on each question type.
3. New metrics: The authors use only 25 questions to validate the new metrics which may be subject to considerable random variation. I believe more comprehensive experiments are needed to validate the effectiveness of the new metrics.
4. It would be beneficial to provide a data statistics comparison with previous datasets.

**Relation To Prior Work:**

Yes

**Summary And Contributions:**

This paper presents MEQA, a benchmark for multi-hop event-centric question answering. MEQA is the first event-centric multi-hop reasoning benchmark which involves both events and entities and presents novel challenges. To enrich diversity, MEQA consists of 5 types of questions: event relation, event bridging, event listing and counting, event comparison, and unanswerable questions. This paper designs a bottom-up process to construct MEQA from the information extraction (IE) dataset. To comprehensively evaluate the performance on this benchmark , this paper also introduces 2 new metrics, namely completeness and logical consistency. This paper conduct extensive experiments, including various prompting methods. The experimental results demonstrate MEQA poses new challenges.

---

> ### Author Rebuttal · Authors · 2024-08-17
>
> Thank you for the comprehensive and insightful feedback! We appreciate your effort in reviewing our paper and are committed to addressing your concerns in the following response.
>
> > 1. New Challenges: Although the authors claim that event-centric multi-hop reasoning is a novel task that presents novel challenges, they do not elaborate in detail. Compared to previous multi-hop reasoning datasets, what specific new challenges does MEQA introduce? Especially if we treat events as ``entities’’.
>
> In Appendix A, we explained the definition and complexity of event-centric questions. It describes the challenges of event-centric questions compared to entity-centric questions.
>
> Briefly, to solve entity-centric questions, we need to reason over entities. In entity-based reasoning chains, nodes exclusively represent entities, with edges denoting relations between these entities. Similarly, we need to reason over events to answer event-centric questions. However, in event-based reasoning chains, nodes may represent entities or events, with edges encompassing relations between entities and the roles entities play in events (heterogeneous graph). For instance, in Figure 2, "kills" serves as the event trigger, signifying an event rather than an entity. Event nodes serve as abstract representations of events, rendering them challenging for current Language Models (LLMs) to process, particularly due to the complexity introduced by their associated edges.
>
> > 2. Question distribution: The distribution of question types is too long-tailed, particularly with event comparison questions accounting for only 0.6%, which translates to roughly 12 questions, which is insufficient for effectively evaluating models. Additionally, it would be better to provide the performance on each question type.
>
> Please refer to our general response.
>
> > 3. New metrics: The authors use only 25 questions to validate the new metrics which may be subject to considerable random variation. I believe more comprehensive experiments are needed to validate the effectiveness of the new metrics.
>
> We conduct a new validation of over 100 questions, similar to other multi-hop QA benchmark papers (e.g. Musique, HotpotQA, 2WikiMultihopQA, etc.). Currently, the correlation scores of the “completeness” and “logical consistency” metrics are 0.693 and 0.601.
>
> > 4. It would be beneficial to provide a data statistics comparison with previous datasets.
>
> The table for data statistics comparison with HotpotQA [1], 2WikiMultihop [2], and Musique [3] is below. Since those benchmark papers do not provide the statistics above, we estimate partial numbers based on the statistics they provide in the paper. Since none of them focus on events, they do not have event-related statistics.
>
> |                      | MEQA  | HotpotQA        | 2WikiMultihopQA | Musique                     |
> | -------------------- | ----- | --------------- | --------------- | --------------------------- |
> | Avg. Question Length | 11.6  | 16              | 12.64           | 15.40 (tokens per question) |
> | Avg. of Hops         | 2.6   | 2               | 2               | 2.4                         |
> | # of Documents       | 211   | 14,810 (golden) | 25,152 (golden) | 7676                        |
> | # of Questions       | 2,093 | 7,405           | 12,576          | 2,459                       |
> | # of Events          | 1565  | -               | -               | -                           |
> | # of Entities        | 2644  | -               | -               | -                           |
>
> Our MEQA is used to benchmark models’ reasoning ability on event-centric questions. We only compare the test set part of the other three benchmarks.
>
> **References:**
> 1. Yang, Zhilin, et al. "HotpotQA: A dataset for diverse, explainable multi-hop question answering." _arXiv preprint arXiv:1809.09600_ (2018).
> 2. Ho, Xanh, et al. "Constructing a multi-hop QA dataset for comprehensive evaluation of reasoning steps." _arXiv preprint arXiv:2011.01060_ (2020).
> 3. Trivedi, Harsh, et al. "♫ MuSiQue: Multihop Questions via Single-hop Question Composition." _Transactions of the Association for Computational Linguistics_ 10 (2022): 539-554.

---

> ### Author Response · Authors · 2024-08-27
> **We want Discussions!**
>
> Dear Reviewer spj2,
>
> We appreciate your efforts in reviewing our paper. With only a few days remaining in the discussion period, please let us know if we have addressed your concerns. We are open to further discussion if needed.

---

> ### Author Response · Authors · 2024-08-29
> **We want discussions!**
>
> Dear Reviewer spj2,
>
> Thank you for your efforts in reviewing our paper. With just two days left in the discussion period, we kindly ask if our responses have addressed your concerns. Please let us know if further clarification is needed.

---

> ### Author Response · Authors · 2024-08-30
> **We Want Discussions!**
>
> Dear Reviewer spj2,
>
> We hope this message finds you well. With only one day remaining in the discussion period, we kindly remind you to review our previous rebuttal contents. Your feedback is invaluable to us, and we would greatly appreciate it if you could confirm whether our revisions have addressed your concerns or if any additional clarifications are needed.

---

### Official Review · Reviewer_sD7f · 2024-08-02
**Useful Benchmark for Event-Centric Question Answering**

**Rating:** 7
**Confidence:** 4
**Correctness:** Yes
**Clarity:** Yes

**Review:**

This paper introduces a challenging benchmark for multi-hop event question answering. Although the question types and hence the reasoning types are fairly limited due to templated generation it still provides a challenging benchmark for testing the reasoning abilities of language models as evidenced by the low performance by SOTA models

**Strengths:**

Sound data collection strategy.

**Additional Feedback:**

None

**Documentation:**

Yes

**Limitations:**

- Missing citations:
https://aclanthology.org/2022.case-1.5/
- Dataset is dominated by two question types, diversify.

**Opportunities For Improvement:**

- Missing citations:
https://aclanthology.org/2022.case-1.5/
- Dataset is dominated by two question types, diversify.

**Relation To Prior Work:**

Missing citations
https://aclanthology.org/2022.case-1.5/

**Summary And Contributions:**

This paper introduces a dataset for Multi-hop Event-centric Question Answering. It consists of 2,093 challenging questions that
8 require a diverse range of complex reasoning over entity-entity, entity-event, and 9 event-event relations; (2) corresponding multi-step QA-format event reasoning 10 chain (explanation) which leads to the answer for each question.

---

> ### Author Rebuttal · Authors · 2024-08-17
>
> Thank you for rating our paper positively! We sincerely appreciate your observant review and insightful suggestions and aim to address your concerns in the following response.
>
> > 1. Missing citations: [https://aclanthology.org/2022.case-1.5/](https://aclanthology.org/2022.case-1.5/)
>
> We will cite this paper in the final version.
>
> > 2. Dataset is dominated by two question types, diversify.
>
> Please refer to our general response.

---

### Official Review · Reviewer_DJfr · 2024-08-03
**Paper on Evaluating LLMs with an Event-Centric Benchmark: comments and suggestions**

**Rating:** 6
**Confidence:** 4
**Correctness:** I think the submission is technically…
**Clarity:** The paper is clear enough.

**Review:**

See my comments regarding Strengths and Weaknesses below.

**Strengths:**

Strengths:
- The focus on evaluating LLMs with challenging tasks is very relevant. Although many benchmarks have been proposed recently, additional ones can be valuable for the research community.
- The authors have carefully designed a method involving human annotators to ensure the highest possible data quality.
- The authors use various metrics to evaluate performance, considering factors such as completeness and logical consistency.

**Additional Feedback:**

-

**Documentation:**

Sufficient details on data collection are provided. The reference github repo is provided as well.

**Ethics:**

No ethical issues detected.

**Limitations:**

I think the limitations should be better addressed (see my comments above).

**Opportunities For Improvement:**

Weaknesses and Limitations:
- The paper only considers ChatGPT (and a fine-tuned version of T5) as the LLM. Including results from other popular LLMs such as Claude, Gemini, and Llama 3 would have been very interesting. This addition would help identify which LLMs perform best on this challenging task. I would recommend the authors consider adding these additional results if possible.
- The starting dataset is WikiEvents, which is post-processed to create the Q&A dataset. However, there is a lack of discussion on potential data leakage since WikiEvents might have been used to train the LLMs (including the KAIROS ontology). Discussing the implications and possible biases would be beneficial.
- The main paper lacks a discussion on related works, with the primary related datasets only mentioned in the Appendix. A detailed discussion of the related datasets in the main paper would help readers understand the contribution better.
- In Table 2, the LLM performance is significantly worse for the proposed dataset compared to others. The authors attribute this to "data quality" (sec. 4.2), but this needs further discussion. Why are other datasets considered lower quality? Providing concrete examples would help clarify this.

Minor Points:
- In Figure 1, adding an example of an entity-centric question would help show the difference between the proposed benchmark and existing ones. More citations of entity-centric benchmarks would also be helpful.
- "After the annotation procedures, the dataset is further split into training, development, and test sets with a proportion of 80%:10%:10%" — Why is this split used if the dataset is only for evaluating LLM performance? Is the training part reserved for the fine-tuned T5?
- "Employing ChatGPT as the foundational method, where each input contains only a context and a question and the output is only an answer (Appendix H), reveals that MEQA presents the greatest challenge" — Specify the ChatGPT version the first time it is mentioned (It is only mentioned later in the text)
- "The accuracy is 88% on average" for humans. In Table 2, the performance is expressed in terms of Precision, Recall, and F1. Consistency in presenting human performance metrics would be helpful, and including human performance in Table 2 would be beneficial.

I hope the authors can address the aformentioned points.

**Relation To Prior Work:**

As mentioned above, I think a better discussion with the prior work is needed in the main paper (and not only in the appendix).

**Summary And Contributions:**

This paper introduces a new dataset designed to benchmark large language models (LLMs) on event-centric questions rather than the traditional entity-based questions. The dataset was created using a semi-automatic approach that includes human annotators to ensure high-quality data. The evaluation shows that this dataset is significantly challenging for SoA LLMs like ChatGPT.

---

> ### Author Rebuttal · Authors · 2024-08-17
>
> Thank you for the comprehensive and insightful feedback! We appreciate your effort in reviewing our paper and are committed to addressing your concerns in the following response.
>
> # Weaknesses and Limitations:
>
> > 1. The paper only considers ChatGPT and T5 as the LLM. Including results from other popular LLMs such as Claude, Gemini, and Llama 3 would be interesting and better identify top performers.
>
> We conduct new experiments on Claude and Llama3. **The results are in Table 1 and 2 in the attached PDF file.** The precision scores of Claude-3 are low mainly because the model failed to follow the answer format and output redundant information in the outputs. In both tables, the trends of all metrics are the same as in Table 1 in the paper. Among GPT-3.5, Claude-3-Haiku, and Llama-3-70b models, GPT-3.5 performs best considering all metrics. We will add these tables in the final version of the paper.
>
> > 2. The starting dataset is WikiEvents, which is post-processed to create the Q&A dataset. However, there is a lack of discussion on potential data leakage since WikiEvents might have been used to train the LLMs (including the KAIROS ontology). Discussing the implications and possible biases would be beneficial.
>
> We mainly test our dataset on GPT-3.5-turbo-1106 and T5 models. GPT-3.5-turbo-1106 was trained on data collected before Sep 2021 [1]. T5 models were released in 2019. However, WikiEvents was released in Oct 2021. Thus, both models should have yet to see the WikiEvent data.
>
> Moreover, even with potential leakages of the original WikiEvent dataset, since the new dataset focuses on reasoning, models do not remember reasoning processes. We design a new experiment to show this. In this experiment, we apply two models, GPT-4-0613 (training data up to Sep 2021 before WikiEvent) and GPT-4-turbo-2024-04-09 (training data up to Dec 2023 after WikiEvent). The two models have similar abilities across other NLP tasks, such as math problems (on GSM8k and MATH), open-domain question answering (on Natual Question, ARC), text summarization, etc, but the GPT-4-turbo-2024-04-09 may be affected by the potential leakage. **The results of the two models are in Table 3 in the attached PDF file.**
>
> Based on the table, the performance of GPT-4-turbo-2024-04-09 does not exceed its of GPT-4-0613. Thus, the potential leakage of WikeEvent has little influence over our MEQA benchmark evaluation on event-based reasoning capabilities. As a similar finding is in Musique, models pre-trained on Wikipedia do not perform well on the multi-hop questions requiring complex reasoning.
>
> > 3. The main paper lacks a discussion on related works, with the primary related datasets only mentioned in the Appendix.
>
> We discuss related works in Appendix B due to the lack of space in the submitted version (It was in 2nd section after the introduction). In the final version, we will move related works to the main pages since one more page will be available. In the paper, “Related Works” (Appendix B) discusses “Multi-hop QA”, “Explainable Complex QA”, and “Event-centric QA”.
>
> > 4. In Table 2, the LLM performance is significantly worse for the proposed dataset compared to others. The authors attribute this to "data quality" (sec. 4.2), but this needs further discussion. Why are other datasets considered lower quality? Providing concrete examples would help clarify this.
>
> We are sorry for the confusion regarding the title name of section 4.2. We will change it to “Data Difficulty Evaluation”. In this section, we did not claim that other datasets were considered lower quality. Results in Table 2 illustrate that our MEQA dataset is more challenging than others.
>
> # Minor Points:
>
> > 1. In Figure 1, adding an example of an entity-centric question would help show the difference between the proposed benchmark and existing ones. More citations of entity-centric benchmarks would also be helpful.
>
> **We will add the entity-centric example in Table 4 in the attached PDF to the paper.** The entity-centric example is from the entity-centric benchmark HotpotQA. Its reasoning type is “entity bridging”. The reasoning starts from “MVP”, hops by “Buddy Hield”, and ends with the answer “Sacramento Kings”. This is different from our example in Figure 1. Our example first hops to an event, then hops to multiple events by event relations, and ends with attributes in those events.
>
> **As a comparison, an event-centric example is also shown in Table 4 in the attached PDF.** To answer the question, models should start reasoning from the Al-Monitor and first locate the reported event; then find all events that happened before the reported event; and finally extract victims in all those events, which are answers to the question.
>
> >2. Why is the split "80%:10%:10%" used if the dataset is only for evaluating LLM performance? Is the training part reserved for the fine-tuned T5?
>
> Yes, the training part is reserved for fine-tuning the T5 model. We split the dataset in this portion to keep the test set the same for all comparison models.
>
> > 3. Specify the ChatGPT version the first time it is mentioned (It is only mentioned later in the text)
>
> We will mention “ChatGPT” as ChatGPT-3.5 in Table 2 and section 4.2, where “ChatGPT” appears first in the paper.
>
> > 4. "The accuracy is 88% on average" for humans. In Table 2, the performance is expressed in terms of Precision, Recall, and F1. Consistency in presenting human performance metrics would be helpful, and including human performance in Table 2 would be beneficial.
>
> The precision, recall, and F1 of human performance on MEQA are added at the end of Table 2 in the paper. **The entire table is shown in Table 5 in the attached PDF.**
>
> **References:**
> 1. https://platform.openai.com/docs/models/gpt-3-5-turbo

---

> > ### Comment · Reviewer_DJfr · 2024-08-17
> > **Thank you to the authors**
> >
> > The authors replied to my comments and doubts. New experiments have been shared. I will raise my score to 6.

---

### Author Rebuttal · Authors · 2024-08-17

We greatly appreciate the detailed feedback from all reviewers. Your insightful suggestions have significantly inspired us to enhance our draft. We are committed to address the reviewers’ concerns by topics. Regarding the common concern raised by the reviewers, we have conducted an additional experiment.

> 1. Dataset is dominated by two question types and the distribution is too long-tailed.

We have added more data for the corresponding “Event Listing and Counting” and “Event Comparison” types. Currently, the proportion of two types of data increase to 6.2% (+50) and 5.0% (+100), respectively. The new distribution nearly matches other question answering datasets, such as HotpotQA (15%, 6%, 2% for the non-dominated reasoning types) and 2WikiMultiHopQA (20%, 4% for the non-dominated reasoning types). In the final released data, we will balance the dataset's distribution and increase both to about 10%.

After adding more data, we also conduct a new experiment to provide the performance of each question type. The performance of each type is tested on the Full Event KG setting. The performance of each type is tested on the Full Event KG setting.

| GPT-3.5-turbo-1106         | Performance |        |        |
| -------------------------- | ----------- | ------ | ------ |
| CoT (Full Event KG)        | Precision   | Recall | F1     |
| Event Relation             | 0.4740      | 0.4492 | 0.4265 |
| Entity Bridging            | 0.5539      | 0.5404 | 0.5094 |
| Event Listing and Counting | 0.3895      | 0.5024 | 0.4049 |
| Event Comparison           | 0.3682      | 0.4622 | 0.3963 |

Unanswerable questions have performance scores since no golden answers or reasoning processes exist.

---

### Comment · Area_Chair_5Jdz · 2024-08-31
**[urgent] Please address the author rebuttals**

Dear reviewers,

Thank you for your hard work.

As the discussion period will end soon (within 12 hours), I strongly encourage reviewers to acknowledge the rebuttals made by the authors, if you have not already done so.

I acknowledge that reviewers already put an effort into making the initial reviews, however, the reviewing process includes discussions within the reviewer group and discussions with authors. Let's make this one last effort to make this reviewing process more productive!

Best,

---

### Decision · Program_Chairs · 2024-09-26

**Decision:**

Accept (Poster)

**Comment:**

Unlike entity-based questions in traditional benchmarks, the proposed dataset MEQA focuses on complex reasoning involving event and entity relations, aimed at evaluating large language models (LLMs) on event-centric multi-hop question answering.

The dataset creation involves a semi-automatic process with human annotation, ensuring high-quality data. Several performance metrics, including completeness and logical consistency, are used for evaluation, and the results show MEQA poses significant challenges for state-of-the-art (SoA) models, such as ChatGPT and T5.

The reviewers agree that this is a challenging benchmark and that the paper is clearly written. The authors have well responded to reviewer concerns such as limited model evaluation (by adding evaluation with Calude and Llama3) and adding more data points, and have persuaded some of the reviewers.